# Identification and Validation of Ferroptosis-Related DNA Methylation Signature for Predicting the Prognosis and Guiding the Treatment in Cutaneous Melanoma

**DOI:** 10.3390/ijms232415677

**Published:** 2022-12-10

**Authors:** Wenna Guo, Xue Wang, Yanna Wang, Shuting Zhu, Rui Zhu, Liucun Zhu

**Affiliations:** 1School of Life Sciences, Zhengzhou University, Zhengzhou 450001, China; 2School of Life Sciences, Shanghai University, Shanghai 200444, China

**Keywords:** cutaneous melanoma, ferroptosis, DNA methylation, prognosis, immune

## Abstract

Cutaneous melanoma (CM) is one of the most aggressive skin tumors with a poor prognosis. Ferroptosis is a newly discovered form of regulated cell death that is closely associated with cancer development and immunotherapy. The aim of this study was to establish and validate a ferroptosis-related gene (FRG) DNA methylation signature to predict the prognosis of CM patients using data from The Cancer Genome Atlas (TCGA) and the Gene Expression Omnibus (GEO) database. A reliable four-FRG DNA methylation prognostic signature was constructed via Cox regression analysis based on TCGA database. Kaplan–Meier analysis showed that patients in the high-risk group tended to have a shorter overall survival (OS) than the low-risk group in both training TCGA and validation GEO cohorts. Time-dependent receiver operating characteristic (ROC) analysis showed the areas under the curve (AUC) at 1, 3, and 5 years were 0.738, 0.730, and 0.770 in TCGA cohort and 0.773, 0.775, and 0.905 in the validation cohort, respectively. Univariate and multivariate Cox regression analyses indicated that the signature was an independent prognostic indicator of OS in patients with CM. Immunogenomic profiling showed the low-risk group of patients had a higher immunophenoscore, and most immune checkpoints were negatively associated with the risk signature. Functional enrichment analysis revealed that immune response and immune-related pathways were enriched in the low-risk group. In conclusion, we established and validated a four-FRG DNA methylation signature that independently predicts prognosis in CM patients. This signature was strongly correlated with the immune landscape, and may serve as a biomarker to guide clinicians in making more precise and personalized treatment decisions for CM patients.

## 1. Introduction

Cutaneous melanoma (CM) is one of the most aggressive malignant skin tumors that arise from mutations in melanocytes [1]. The incidence of CM has increased rapidly over the past decades [2,3]. Although medical advances have been able to alleviate symptoms and reduce the burden of tumors, they still provide limited help in improving survival, with 5-year overall survival (OS) of 15% [4,5]. An accurate assessment of CM progression is essential to improve OS [6]. Therefore, there is a need to identify novel robust biomarkers to predict patient prognosis.

Excessive ultraviolet (UV) radiation is an important environmental trigger in the pathogenesis of CM [7]. UV from sunlight can lead to excessive intracellular production of reactive oxygen species (ROS), causing oxidative stress damage to cells [8]. Mitochondria, as a major iron-rich and ROS-producing organelle, is considered to be an important site of cell ferroptosis [9], and the accumulation of ROS is one of the characteristics of ferroptosis [10]. Thus, the role of ferroptosis-related genes (FRGs) in the development and prognosis of CM has attracted our interest. Ferroptosis is a newly discovered form of programmed cell death caused by iron-dependent lipid peroxidation [11]. Ferroptosis activity evaluated by ferroptosis-related gene (FRG) expression is effective in tumorigenesis and therapy [12,13]. There is growing evidence that FRGs are closely associated with tumor cell proliferation, invasion, metastasis, apoptosis, and tumor therapeutic response in a variety of cancers, including CM [14,15]. For example, Erastin is a ferroptosis inducer that significantly enhances BRAF inhibitor-induced CM cell death [16]. miR-137 enhanced the therapeutic efficacy by increasing CM ferroptosis [17]. Epigenetic modifications play an important role in tumorigenesis. DNA methylation is a well-researched form of epigenetic modification that has been demonstrated to regulate tumorigenesis, progression, and treatment in a variety of tumors including CM [18]. DNA methylation can regulate ferroptosis by modulating the transcription of corresponding genes in tumors, affecting various biological behaviors and alterations in signaling pathways in tumor cells [19]. Furthermore, due to its high stability, frequency, and accessibility, DNA methylation is widely used as a diagnostic, predictive, and prognostic biomarker for various cancers [20,21]. It has also been found that biomarkers of DNA methylation have higher performance compared to gene expression [22,23]. Several methylation biomarkers have been used as an effective tool to predict prognosis [20,24,25]. However, the prognostic potential and underlying molecular mechanisms of ferroptosis-related DNA methylation for CM patients are unknown.

Therefore, we collected DNA methylomics, transcriptomics, and clinical data of CM patients from The Cancer Genome Atlas (TCGA) and the Gene Expression Omnibus (GEO) databases. Ferroptosis-related DNA methylation sites associated with patient prognosis were screened out by univariate Cox regression, and a ferroptosis-related DNA methylation signature was identified as a prognostic biomarker by multivariate Cox regression analysis based on the TCGA CM cohort. The GEO cohort was used as an independent dataset to validate the signature. The independence of the signature and the cohort’s impact on the tumor immune microenvironment were investigated. Functional enrichment analysis was used to explore functional differences between high- and low-risk groups. This study contributes to uncovering the possible mechanism of FRG DNA methylation on CM development and prognosis, providing new research targets for the prognostic prediction and the treatment of CM.

## 2. Results

### 2.1. Screening and Establishment of Prognosis-Related FRG DNA Methylation Signature

A total of 567 FRGs and corresponding 8781 DNA methylation sites of these FRGs were extracted. Through univariate Cox regression analysis, 631 DNA methylation sites that were significantly associated with patient prognosis were screened out (*p* < 0.01). These DNA methylation sites were subsequently used in multivariate Cox regression analyses to construct a prognostic signature. An optimum model consisting of four DNA methylation sites (cg12336709, cg23750391, cg15674193, and cg06904403) was constructed for predicting OS. Forest plots showed the association of these four DNA methylation sites with patient OS in a univariate and multivariate Cox proportional hazard model (Figure 1). The genes corresponding to these four sites were *ZEB1* (zinc finger E-box binding homeobox 1), *CISD1* (CDGSH iron sulfur domain 1), *LRRFIP1* (LRR binding FLII interacting protein 1), and *DUOX1* (dual oxidase 1). The chromosomal locations of these four DNA methylation sites and their *p* values in Cox regression analysis are shown in Table 1.

### 2.2. Evaluation of the FRG DNA Methylation Prognosis Signature

The risk scores of CM patients were calculated according to the riskScore formula: riskScore = (−1.959 × β value of cg12336709) + (−9.761 × β value of cg2375039) + (−1.229 × β value of cg15674193) + (1.418 × β value of cg06904403). Patients were separated into high- and low-risk groups based on the median risk score (−2.137) (Figure 2A), and the number of deaths increased with the increasing risk score (Figure 2B). Principal component analysis (PCA) showed that the contribution of the four components was sustained and there were identifiable dimensions between the high- and low-risk groups (Figure 2C). In addition, there was a significant difference in DNA methylation levels between patients with long-term (>5 years) and short-term (<5 years) survival (Figure 2D) (*p* < 0.05, Mann–Whitney U test).

Kaplan–Meier survival analysis showed significantly lower survival probability in the high-risk group (*p* = 1.19 × 10^−12^) (Figure 3A). ROC curve analysis indicated good predictive accuracy for the four-DNA methylation signature, with area under the curve (AUC) values of 0.738 (95% CI: 0.66–0.82), 0.730 (95% CI: 0.68–0.79), and 0.770 (95% CI: 0.72–0.82) for the 1-year, 3-year, and 5-year survival, respectively (Figure 3B). To further investigate the prognostic value of this signature, we analyzed another independent DNA methylation dataset of CM patients. Patients were divided into high- and low-risk groups according to the median risk score (−2.137) obtained from the training TCGA cohort. As expected, survival prognosis was significantly poorer in patients with high-risk scores (*p* = 0.0052) (Figure 3C), with AUCs of 0.773 at 1 year, 0.775 at 3 years, and 0.905 at 5 years (Figure 3D), confirming the good prognostic predictive role of the signature for prognosis assessment of CM patients.

### 2.3. Correlation between FRG DNA Methylation Signature and Clinical Characteristics

We further analyzed the value of the signature in patients stratified by different clinical factors in TCGA, and investigated whether clinical characteristics could distinguish patients’ prognostic survival. The results demonstrated that male patients had higher risk scores and poorer OS (Figure 4A), whereas there was no significant difference between patients stratified by gender in survival analysis (Figure 4E). Patients who were older (≥60) or with thicker Breslow depth (≥2 mm) had significantly higher risk scores and worse OS (Figure 4B,C,F,G). Stage II patients tended to have higher risk scores than stage I patients (Figure 4D), and stratified survival analysis indicated that patients with advanced stage (stages III and IV) had poorer OS (Figure 4J). Pathogenic variation of BRAF and NRAS genes plays a very important role in CM, about 50% of CM carry an activating *BRAF* mutation [26]. Here, we analyzed the alterations of *NRAS* and *BRAF* in high- and low-groups, and found that patients in the low-risk group had more *BRAF* mutations and fewer *NRAS* mutations. *BRAF*-mutant patients had lower risk scores than *BRAF* wild-type (*BRAF*-wt) patients (Figure 4F), but the survival analysis showed no significant difference between *BRAF*-mutant and *BRAF*-wt groups (Figure 4L). For the *NRAS* gene, there were no significant differences in risk scores or survival analyses between the *NRAS-*mutant and *NRAS-*wt groups (Figure 4E,K).

To explore the independence of the signature from other clinical factors (gender, age, stage, Breslow depth, and *NRAS* and *BRAF* variations), we performed univariate and multivariate Cox regression analyses (Figure 5A,B). The results showed that risk score, age, and stage were correlated with the OS of CM patients in univariate Cox regression, and remained significant in the multivariate Cox regression analysis, suggesting that the signature could be regarded as an independent prognostic indicator. Subsequently, we developed a nomogram for OS prediction using the risk score, age, and stage (Figure 5C). The results indicated that the predictive nomogram for OS was useful for clinical management.

### 2.4. Association of FRG DNA Methylation Prognostic Signature with Immune Landscape

To further understand the potential correlation of the signature with the immune landscape of the CM samples, we compared the differences in the various immune cell components between the low- and high-risk groups. The results indicated that risk scores were negatively correlated with ESTIMATE scores, immune scores, and stromal scores (Figure 6A). The ESTIMATE score, immune score, and interstitial score were significantly increased in the low-risk group (Figure 6B–D). TME immune cell infiltration analysis showed that high-risk score was negatively associated with plasma cells, CD8+ T cells, activated CD4 memory T cells, Gamma Delta T cells, and M1 macrophages, while it was significantly positively correlated with follicular helper T cells, resting NK cell, M2 macrophages, and activated mast cell (Figure 6A,E).

Considering the importance of checkpoint inhibitors in clinical treatment, we further compared the differences in the expression of 12 ICBs in the high- and low-risk groups and found substantial differences in PD-1, PD-L1, PD-L2, CTLA4, CD276, CD80, IDO1, CD4, CD8A, CD8B, and CD86 between the two groups (Figure 7). These results indicated that the low-risk group exhibited a higher immune infiltration status and that immune checkpoint-related pathways might play an essential role in the good prognosis of the low-risk group. This signature could help to select the appropriate checkpoint inhibitors to treat CM patients. Finally, considering that immune infiltration is an important parameter for CM patients, a nomogram was constructed to predict patients’ OS based on risk score, age, stage, and immune score (Figure 8).

### 2.5. Functional Enrichment Analyses

To investigate the underlying mechanism in different prognostic risk groups, we identified the DEGs between the high- and low-risk groups. A total of 716 up-regulated and 76 down-regulated genes were identified in the low-risk group (Figure 9A). Thereafter, GO, KEGG, and Reactome analyses were performed on these DEGs. GO enrichment analysis demonstrated that the up-regulated genes in the low-risk group were mainly associated with immune response, immunological synapse, T cell activation, and positive regulation of T cell proliferation (Figure 9B and Appendix A). KEGG pathway enrichment analysis revealed that immune-related pathways such as cytokine-cytokine receptor interaction, chemokine signaling pathway, NF-kappa B signaling pathway, and B cell receptor signaling pathway were significant (Figure 9C and Appendix A). Reactome analysis showed that these up-regulated genes were mainly enriched in the immune system, cytokine signaling in the immune system, the adaptive immune system, and FCGR activation (Figure 9D and Appendix A). The above results illustrated that the risk signature might be involved in immune microenvironment formation in CM patients

## 3. Discussion

CM is a significant cause of skin cancer-related death worldwide with a poor prognosis [3,27]. Despite advances in CM treatment with targeted therapies combined with immunotherapy in recent decades, cross-drug resistance still affects the prognosis of patients. Well-established risk assessment and treatment stratification tools can help to select treatment options and improve patients’ outcomes [28]. In this study, we combined ferroptosis-related gene (FRG) sets and other datasets to construct and validate a ferroptosis-related DNA methylation signature in CM. The signature may be the first risk-prediction signature based on FRGs DNA methylation with good clinical applicability in CM. 

Ferroptosis refers to an iron-dependent, non-apoptotic form of cell death that plays an essential role in the tumor microenvironment (TME) [11,29]. Dysregulation of ferroptosis has been associated with the development and prognosis of several cancers [12,30]. Among the genes corresponding to DNA methylation in the prognostic signature, we identified that *ZEB1* and *DUOX1* are drivers of ferroptosis and that *CISD1* is a suppressor of ferroptosis, which play a crucial role in cancer development [31,32]. For example, ZEB1 is a major element in the control of epithelial-to-mesenchymal transition (EMT) and is closely related to ferroptosis sensitivity in cancer cells, which can influence tumorigenesis from the early steps of cancer [32,33,34]. Moreover, ZEB1 promotes immune escape in melanoma [35]. *CISD1*, a negative regulator of autophagy, was differentially expressed in a variety of tumors and correlated with OS [36,37]. Meanwhile, *CISD1* can be served as a prognostic biomarker in patients with hepatocellular carcinoma [38]. *DUOX1* is a new target for macrophage reprogramming, and is involved in the redox regulation of EGFR signaling; ATP-mediated *DUOX1* activation led to the activation of the NF-kappaB pathway and production of IL-8 [39,40,41]. *LRRFIP1* plays an important role in the invasion of tumor cells [42]. High expression of *LRRFIP1* was found to be associated with a better response to teniposide in glioblastoma, and could be a candidate gene for tumor-targeted therapy [43]. The major biological characteristics of these four genes are summarized in Table 2. Here, we found the signature consisting of DNA methylation of these four genes was closely correlated with the prognosis of CM patients. 

Aberrant DNA methylation is associated with tumorigenesis of diverse malignancies, including CM [44,45]. Epigenetic changes have been shown to alter gene expression and play a crucial role in the occurrence and progression of cancer [46]. For example, *PD-L1* and *PD-L2* promoter methylation can regulate their mRNA expression and are associated with patient OS in CM [22]. Here, correlation analysis between DNA methylation levels and gene expression indicated that DNA methylation levels at the cg12336709 site were significantly positively correlated with *ZEB1* expression (*p* < 0.0001), while the expression of *CISD1* and *DUOX1* was significantly negatively correlated with their methylation levels (Appendix A), suggesting that these DNA methylations may regulate gene expression and, thus, affect CM development.

Tumor immunotherapy is a new therapeutic approach that has made substantial progress in both fundamental research and clinical practice in recent decades [47]. Here, we calculated immune and stromal scores of tumor samples to investigate the differences in immune microenvironment between low- and high-risk groups, and found that ESTIMATE scores, immune scores, and stromal scores were significantly higher in the low-risk group. We also evaluated the relationship between the infiltration of immune cell factors with the risk score, and found that CD8 T immune cells and activated CD4 memory T cells were negatively correlated with the risk score, while M2 macrophages and activated mast cells were positively correlated with the risk score. High fractions of activated memory CD4 T cells were known to have an anti-tumor function [48], while high fractions of activated mast cells and M2 macrophage have a pro-tumorigenic effect, and are associated with poor prognosis [49,50]. Meanwhile, most immune checkpoint genes also presented strong activity in the low-risk group, such as *PD-1*, *PD-L1*, *PD-L2*, *CTLA4*, *CD4*, *CD8A*, *CD8B,* and *IDO1*. Moreover, functional enrichment analysis demonstrated that the up-regulated genes in the low-risk group were mainly associated with immune response, immunological synapse, T cell activation, positive regulation of T cell proliferation, cytokine–cytokine receptor interaction, the chemokine signaling pathway, the NF-kappa B signaling pathway, and the B cell receptor signaling pathway, the immune system, cytokine signaling in the immune system, the adaptive immune system, and FCGR activation. The up-regulated genes in the low-risk group may be able to activate immune responses. Patients with higher immune activity have a better prognosis [51]. The inhibition of NF-kappa B was correlated with increased disease-free survival in patients with breast cancer [52]. These findings indicated that the FRG DNA methylation signature might influence the prognosis of CM patients through modulating the immune microenvironment of CM.

In addition, univariate Cox regression, Kaplan–Meier, and ROC analyses were carried out on the four individual methylation sites to further analyze their prognostic significance in CM. Kaplan–Meier analysis showed that the single DNA methylation site also could distinguish patients between high- and low-risk, and patients with higher methylation levels for cg12336709, cg23750391, and cg15674193 exhibited higher OS (Appendix A), and patients with higher cg06904403 methylation had lower OS (Appendix A). ROC analysis showed that the predictive performances of individual methylation sites were lower than the combination of these four DNA methylation sites (Appendix A). These results implied a combination of methylation sites can provide a better prognostic prediction for CM patients.

## 4. Materials and Methods

### 4.1. Dataset Source and Pre-Processing

Publicly available DNA methylation data and clinical information of CM patients were downloaded from TCGA “https://portal.gdc.cancer.gov/ (accessed on 26 September 2021)” and GEO databases “https://ncbi.nlm.nih.gov/geo/query/acc.cgi?acc=GSE51547 (accessed on 26 September 2021)”. The somatic mutation data were obtained from TCGA database, and *NRAS* and *BRAF* mutation information was extracted. DNA methylation levels were expressed as β values ranging from 0 (no methylation) to 1 (100% methylation). FRGs were acquired from the FerrDb database “http://www.zhounan.org/ferrdb/ (accessed on 17 September 2021)” [53], and the corresponding DNA methylation site information was extracted using Perl script. Patients from TCGA database were utilized as the training cohort for the identification and construction of the prognostic signature, and other data obtained from the GEO database were used as an independent validation set validation of prognostic signature. 

### 4.2. Prognosis-Related DNA Methylation Filtering and Risk Model Construction

The FRG DNA methylation levels in TCGA data were subjected to univariate Cox regression analysis to determine DNA methylation related to OS. Statistically significant (*p* < 0.01) variables were then selected for multivariate Cox regression analysis to construct models comprising all possible combinations of two to five factors, aiming at further filtering out combined biomarkers associated with OS. Hazard ratios (HRs) and 95% confidence intervals (CIs) were calculated. The prognostic risk scores were calculated for each patient by summing the DNA methylation levels and the corresponding regression coefficients. Patients were divided into high- and low-risk groups based on the median risk score. 

### 4.3. DNA Methylation Prognostic Signature Validation and Nomogram Construction 

To assess the usability of the prognostic signature, the Kaplan–Meier survival curve and Wilcoxon rank test were used to demonstrate and compare the difference in prognosis between the two groups [54]. ROC curves were applied to calculate the sensitivity and specificity of the signature and to estimate the prognostic performance [55]. Univariate and multivariate Cox regression analyses were performed to assess the independence of the signature. Nomograms for the 1-, 3-, and 5-year OS were constructed to predict survival rates by summing the points corresponding to each factor of patients.

### 4.4. Immunogenomic Landscape Evaluation 

The infiltration levels of stromal and immune cells were calculated for each CM patient using the “ESTIMATE” R package [56]. Three kinds of immunophenoscores and infiltration of 22 immune cells for CM patients were obtained, and the differences between the high- and low-risk groups were compared. We also analyzed the correlation of FRG DNA methylation signature with immune checkpoint-related genes (ICGs), and compared the expression values of ICGs in patients between high- and low-risk groups. Correlations were calculated using Spearman correlation analysis, and the differences between the two groups were evaluated using Mann–Whitney U-test.

### 4.5. Functional Enrichment Analysis

Differentially expressed genes (DEGs) between low- and high-risk patients were determined using R “limma” package, and the adjusted *p* < 0.01, |logFC| ≥ 0.5 were considered a statistically significant difference. GO functional enrichment and KEGG pathway analysis were conducted and the biological function and signaling pathways were demonstrated by ggplot2.

### 4.6. Survival Analysis

Statistical analyses were performed using R language (version 4.0.3). Univariate and multivariate Cox proportional hazards analyses were performed using the R “survival” package. Time-dependent ROC analysis was conducted using R “survivalROC” package, and ROC curves were plotted for 1-, 3-, and 5-year. Nomograms were established with the R “rms” package. The differences between survival curves were detected using the generalized Wilcoxon test. In addition, the correlation between DNA methylation sites and the relative expression of the corresponding genes were examined by Spearman correlation analysis. Unless otherwise stated, *p* < 0.05 was considered significant.

## 5. Conclusions

In summary, we constructed and validated a ferroptosis-related four-DNA methylation signature that could be utilized as an independent and effective biomarker to predict the prognosis of CM patients. The signature was correlated with tumor immune infiltration and immune checkpoint genes, and would facilitate the selection of more effective immunotherapeutic strategies. Functional enrichment analysis revealed that the signature might be involved in immune microenvironment formation in CM patients. Despite the specific function requiring further exploration, this study provided a theoretical foundation for improving the clinical treatment and promoting the development of personalized immunotherapy and precision medicine for CM.

## Figures and Tables

**Figure 1 ijms-23-15677-f001:**
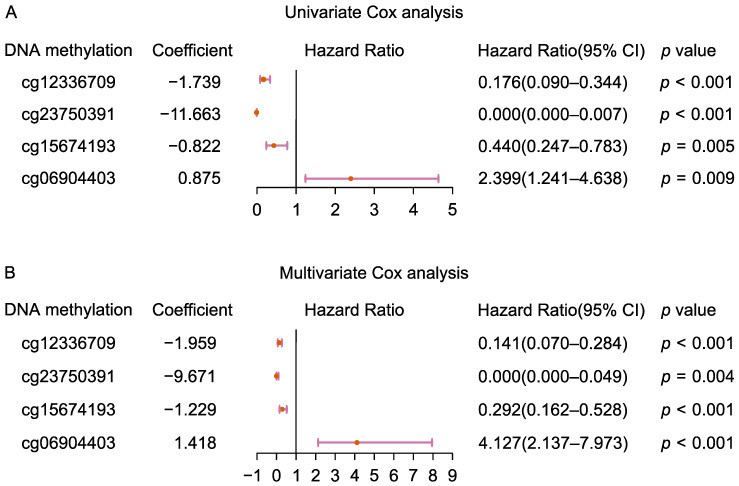
The forest plots of univariate and multivariate cox regression about four FRG DNA methylation site in the TCGA cohort. (**A**) Forest plots of univariate Cox regression analysis using single methylation levels at four DNA methylation sites as variables, respectively. (**B**) Forest plots of Multivariate Cox regression analysis using methylation levels at four DNA methylation sites as covariates. The forest plot was generated to show the connection between four FRG DNA methylation sites and the OS of CM patients.

**Figure 2 ijms-23-15677-f002:**
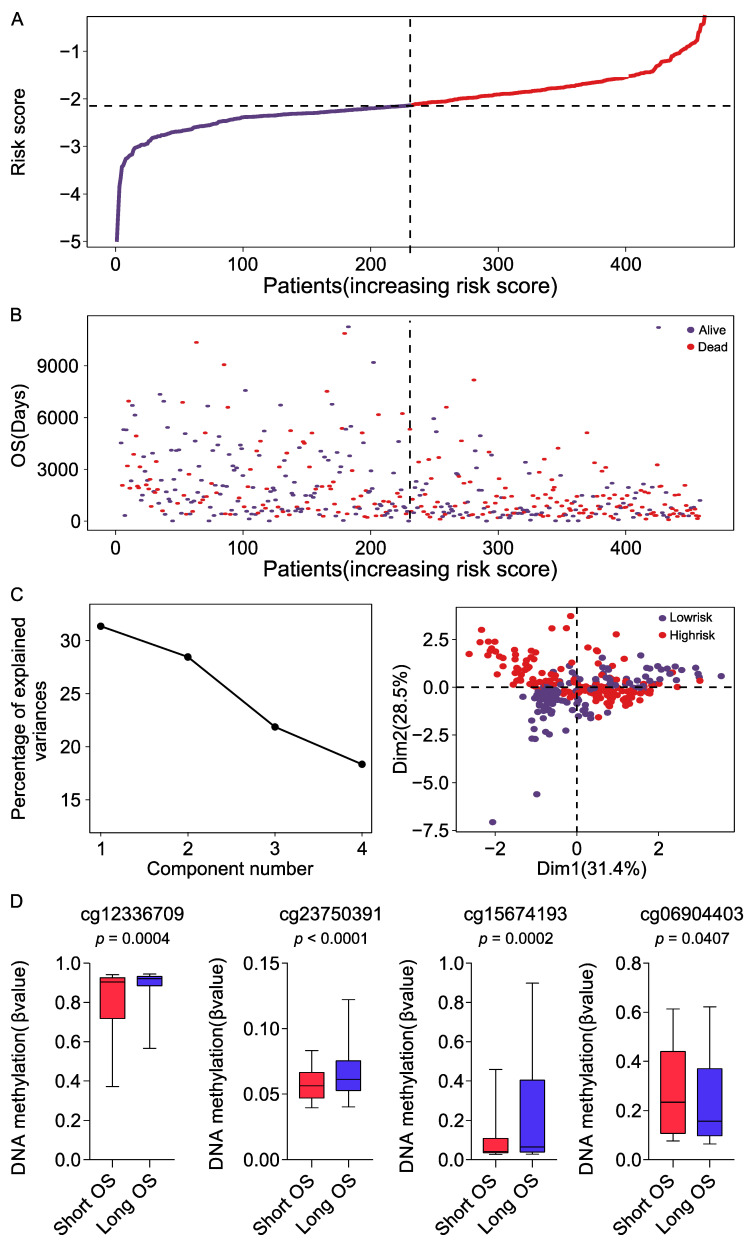
Correlation between prognostic signature and overall survival (OS) of patients in the TCGA cohort: (**A**) The distribution of risk score (upper) and OS (bottom). Patients were divided into high- and low-risk groups using the median score as the cut-off value. (**B**) Survival status of each patient as risk score increases. (**C**) Principal component analysis (PCA) of four DNA methylation sites. The contribution of each component was ranked according to the magnitude of the corresponding percentage of explained variants in the PCA (left). PCA plots showed the high- and low-risk groups in TCGA database (right). (**D**) DNA methylation levels of four FRG sites in patients with short (OS < 5 years) and long survival (OS > 5 years). The thick line represents the median, and the bottom and top of the box are the 25th and 75th percentiles (interquartile range), respectively, and the whiskers indicated the 5th and 95th percentiles. Differences between short and long survival groups were compared by the Mann–Whitney U test, and *p* values are shown below the plots.

**Figure 3 ijms-23-15677-f003:**
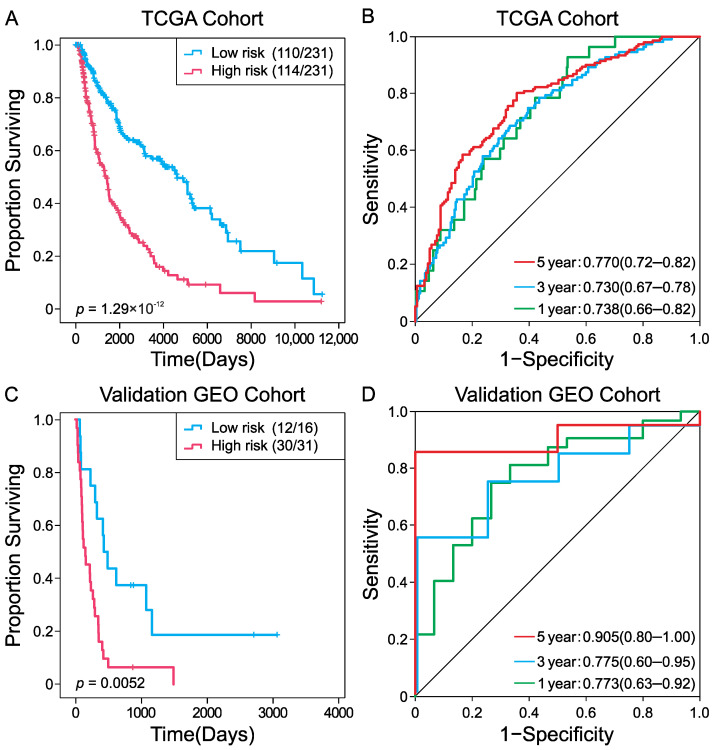
Kaplan–Meier survival and ROC analyses of the four-FRG DNA methylation prognostic signature in the discovery TCGA cohort (**A**,**B**) and the validation GEO cohort (**C**,**D**). Kaplan–Meier analysis combined with Wilcoxon test was used to visualize and compare the OS in low- and high-risk groups. AUC values for the 1, 3, and 5-year survival to show the prognostic performance of the signature.

**Figure 4 ijms-23-15677-f004:**
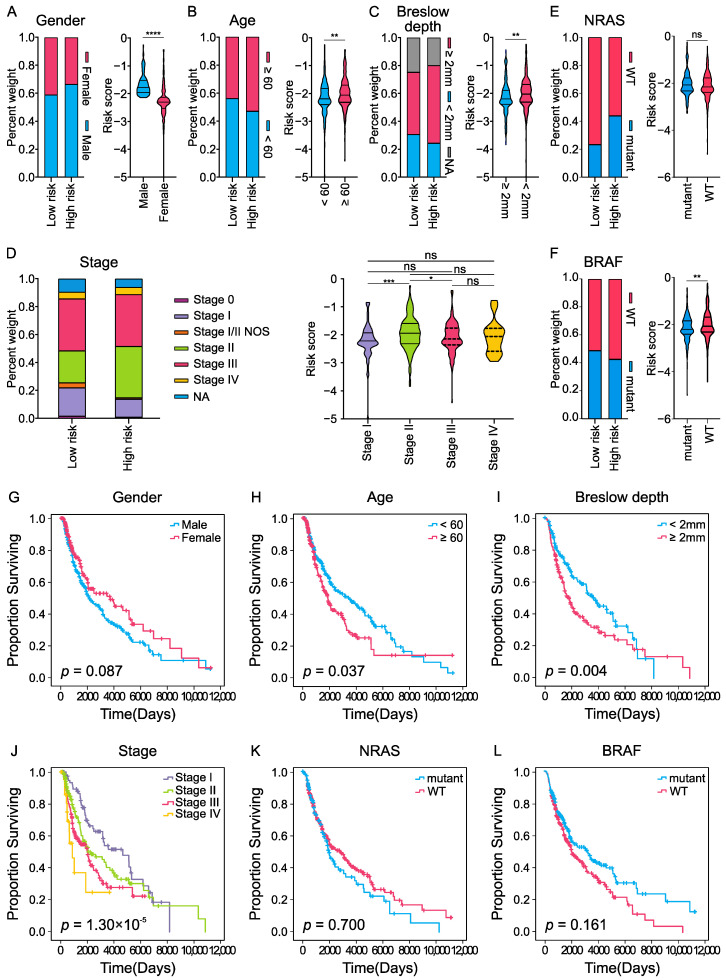
Association between the four-FRG DNA methylation signature and clinical characteristics in the TCGA cohort: (**A**–**F**) Proportion of clinical characteristics (age, gender, stage, and Breslow depth) in the low- or high-risk group, and the distribution of risk scores in different groups based on clinical characteristics. **** *p* < 0.0001, *** *p* < 0.001, ** *p* < 0.01, * *p* < 0.05, ns means no significance. (**G**–**L**) Kaplan–Meier analysis of CM patients stratified by clinical characteristics.

**Figure 5 ijms-23-15677-f005:**
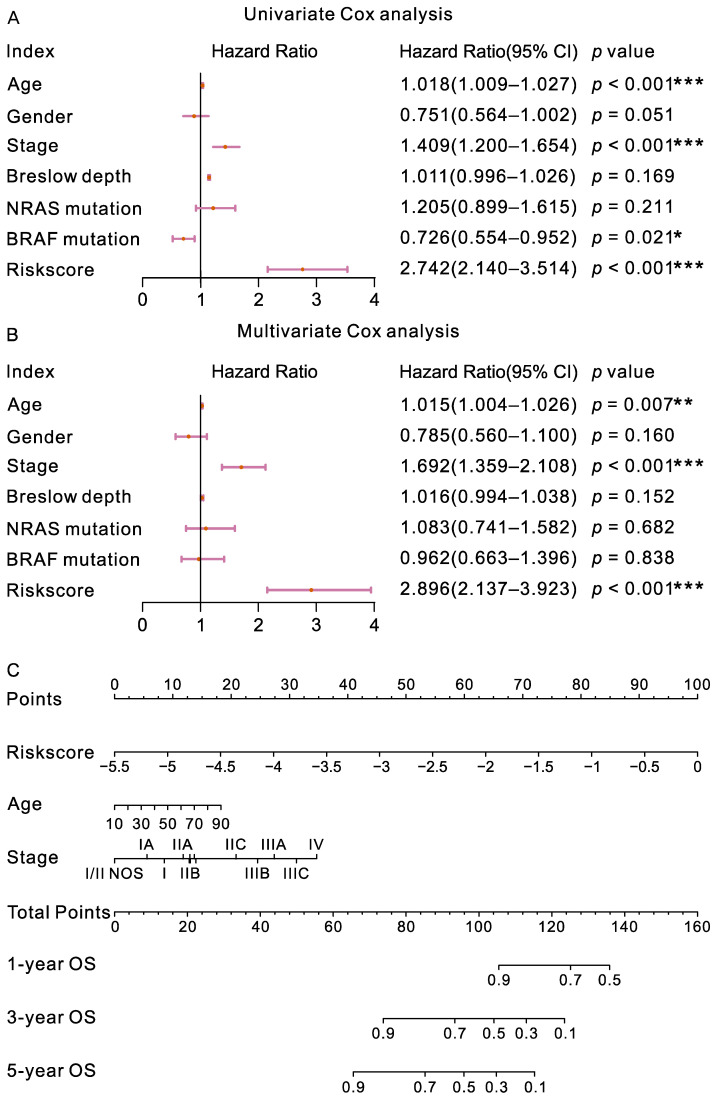
Univariate (**A**) and multivariate (**B**) Cox regression of risk scores and clinical characteristics. *** *p* < 0.001, ** *p* < 0.01, * *p* < 0.05. (**A**) Forest plots of univariate Cox regression analysis using risk scores and clinical characteristics as variables, respectively. (**B**) Forest plots of Multivariate Cox regression analysis using risk scores and clinical characteristics as covariates. The forest plot was generated to show the connection between risk scores, clinical characteristics and the OS of CM patients. (**C**) Nomogram for the prediction of OS in CM patients. The nomogram consisted of age, clinical stage, and the risk score.

**Figure 6 ijms-23-15677-f006:**
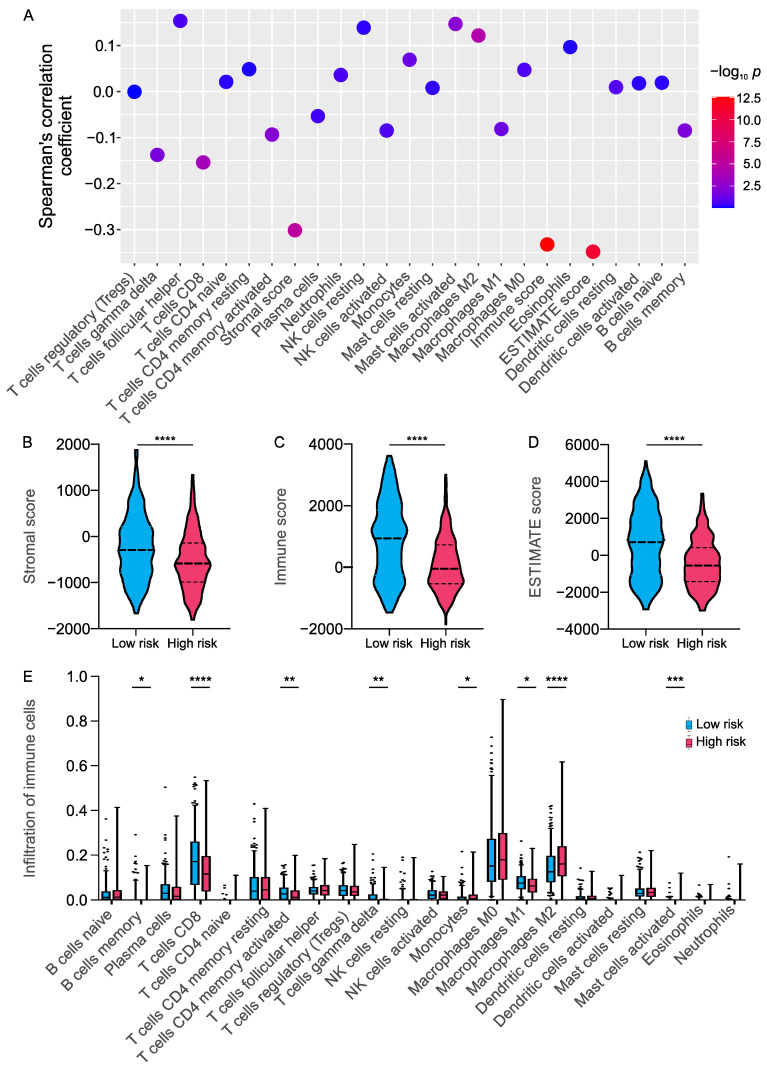
The differences in immune infiltration between high- and low-risk CM patients: (**A**) Correlation analyses. The color of the dot indicates the level of significance (−log_10_
*p*). (**B**–**E**) Violin plot depicted the differences of the stromal score, immune score, ESTIMATE score and the infiltration of 22 immune cells between patients in high- and low-risk groups. **** *p* < 0.0001, *** *p* < 0.001, ** *p* < 0.01, * *p* < 0.05.

**Figure 7 ijms-23-15677-f007:**
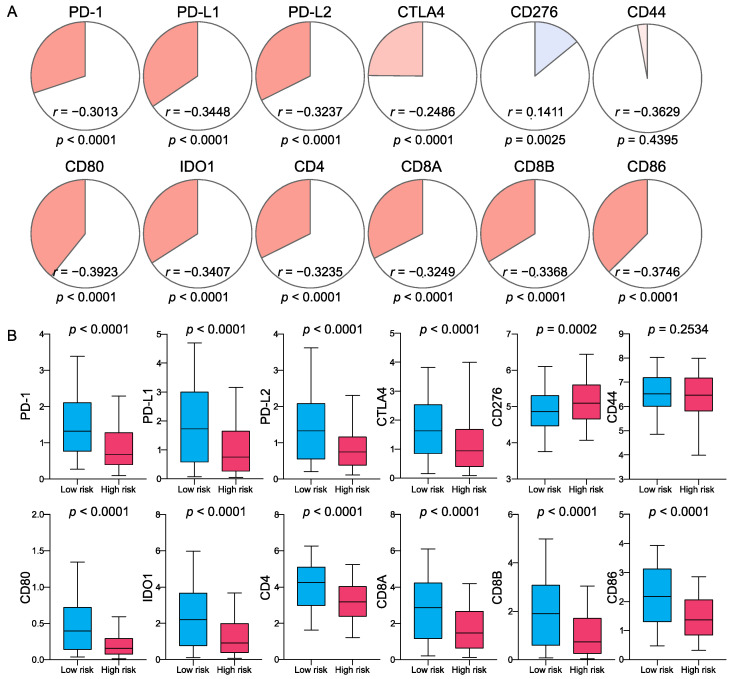
Correlation of FRG DNA methylation signature with immune checkpoint-related genes (ICGs): (**A**) Correlation analysis. The circle symbols represent Spearman’s correlation coefficients, and *p*-values are shown below. Blue indicates positive correlation and red indicates negative correlation. (**B**) Differences in the expressions of ICGs between low- and high-risk groups.

**Figure 8 ijms-23-15677-f008:**
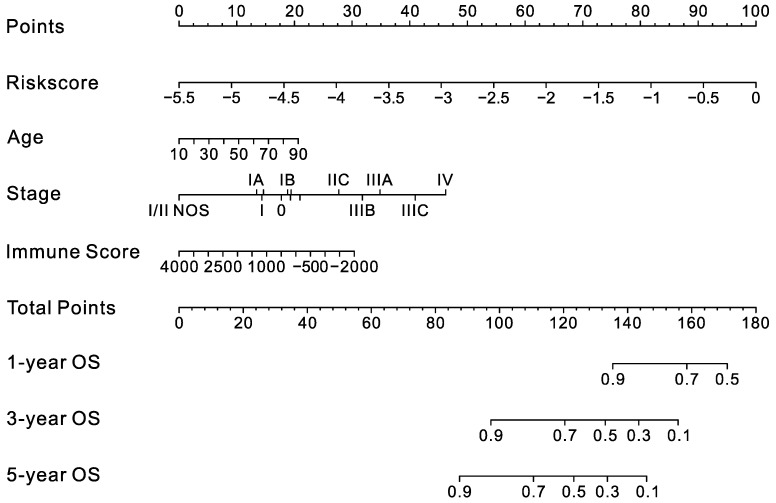
Nomogram for the prediction of OS in CM patients. The nomogram consisted of age, clinical stage, immune score, and the risk score.

**Figure 9 ijms-23-15677-f009:**
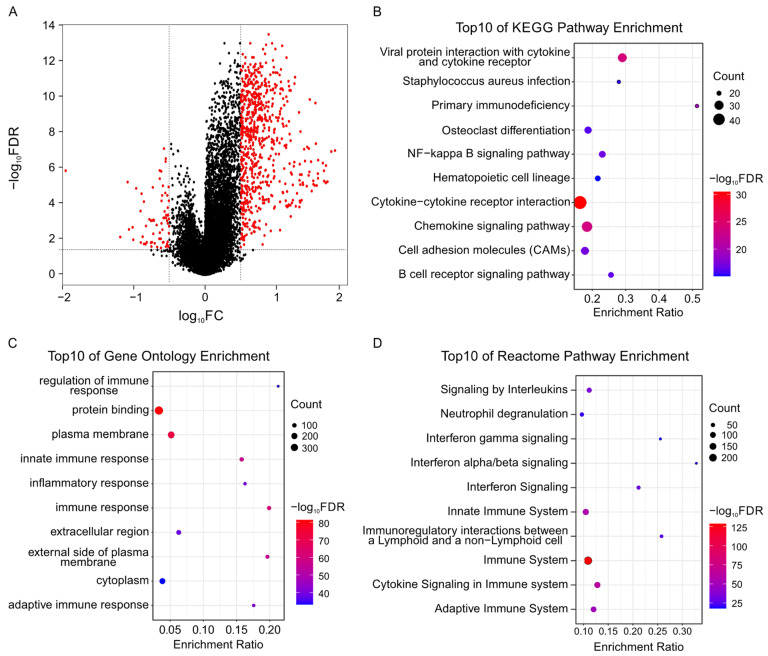
Functional enrichment analysis of differentially expressed genes (DEGs) between the high- and low-risk groups: (**A**) Volcano plot of DEGs. The red plots represent significantly differentially expressed genes, the black plots represent nonsignificant genes. (**B**) Top 50 GO terms of differentially up-regulated genes in low-risk group. (**C**) Top 50 KEGG pathway terms of differentially up-regulated genes. (**D**) Top 50 Reactome terms of differentially up-regulated genes. The size of the dot represents the number of genes and the color indicates the level of significance (−log_10_FDR). The FDR value is the adjusted *p* value.

**Table 1 ijms-23-15677-t001:** Four significantly survival-related FRG DNA methylation sites in TCGA dataset.

Probe ID	Chromosomal Location	Gene Symbol	CGI Coordinate	Feature Type	*p* Value ^1^	*p* Value ^2^
cg12336709	chr10: 31414038–31414039	ZEB1	chr10:31318298–3132116	NA	3.98 × 10^−7^	4.15 × 10^−8^
cg23750391	chr10: 58268743–58268744	CISD1	chr10:58267247–58269481	Island	7.04 × 10^−4^	0.0044
cg15674193	chr2: 237627267–237627268	LRRFIP1	chr2:237626918–237628197	Island	0.0052	4.40 × 10^−5^
cg06904403	chr15: 45129380–45129381	DUOX1	chr15:45129038–45130196	Island	0.0092	2.50 × 10^−5^

^1^. in univariate Cox regression analysis; ^2^. in multivariate Cox regression analysis.

**Table 2 ijms-23-15677-t002:** Major biological characteristics of four genes corresponding to four DNA methylation sites in the FRG DNA methylation signature.

Gene Symbol	Official Full Name	Relation to Ferroptosis	mRNA Expression in CM	Function
ZEB1	Zinc finger E-box binding homeobox	Driver	Down-regulated	transcriptional repression of interleukin 2; control epithelial-to-mesenchymal transition (EMT); promotes immune escape.
CISD1	CDGSH iron sulfur domain 1	Suppressor	Up-regulated	regulate autophagy and oxidation.
LRRFIP1	LRR binding FLII interacting protein 1	Unclassified	Down-regulated	epigenetically regulated gene; plays a role in the invasion and metastasis of tumor cells.
DUOX1	Dual oxidase 1	Driver	Down-regulated	known as dual oxidase; target for macrophage reprogramming; plays a role in the activity of thyroid peroxidase, lactoperoxidase, and in lactoperoxidase-mediated antimicrobial defense at mucosal surfaces.

## Data Availability

Data presented in this study are contained within this article and in the Appendix A, or are available upon request to the corresponding author.

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
