# Peer review of "Identification and Validation of Ferroptosis-Related DNA Methylation Signature for Predicting the Prognosis and Guiding the Treatment in Cutaneous Melanoma"

_ijms, 2022, doi:10.3390/ijms232415677_

Round 1

Reviewer 1 Report

This article describes the possibility of ferroptosis-related genes (FRGs) methylation signature as a prognostic biomarker to make the personalized treatment for cutaneous melanoma (CM) patients by analyzing the correlation between (FRGs) methylation signature and CM status.

The experimental design seems to be exquisite, and the results obtained in each experiment are consistent.

  However, the following points would be considered to improve this article.

Minor point

1)   It does not seem to describe the reason why authors focused ferroptosis and/or

ferroptosis-related genes methylation to analyze the CM status (or the relationship

between CM and ferroptosis) in “Introduction”.

Dose the melanoma cells show ferroptosis drastically, or show more drastic than other cancer cells ?

2) Line 262-263 : “longer” and “lower” are not responding properly.

                longer higher (?)

3) Insufficient description

In “References” section

  Line 375 : 10(9). 10(9): p.2320.

  Line 390 : 13(24). 13(24): p.6217.

  Line 394 : 8. 8: e44310.

  Line 398 : 2021. 2022. 86(2): p.312-321.

  Line 399 : 13(12). 13(12): p.2875.

  Line 404 : 478(2): p.838-44. 478(2): p.838-844.

  Line 410 : 10(3). 10(3): e003484.

  Line 419 : 284(26): p.17858-67. 284(26): p.17858-17867.

  Line 420 : 8(1). 8(1): e000622.

  Line 422 : 365(1): p.132-40. 365(1): p.132-140.

  Line 424 : 446(4): p.1261-7. 446(4): p.1261-1267.

  Line 436 : 66(2): p.605-12. 66(2): p.605-612.1

  Line 442 : 56(2): p.337-44. 56(2): p.337-344.

3) There is an extra description.

   a) Line 22 : higher higher immunophenoscore

                → higher immunophenoscore

[Please delete “higher”]

Author Response

We thank you very much for giving us an opportunity to revise our manuscript and we appreciate you very much for your comments and suggestions on our manuscript. We have carefully read the reviewer’ comments and tried our best to revise our manuscript according to the reviewer’s suggestions. The detailed explanation of revision in response to the reviewers’ concerns is given point by point in the following pages, and the part we have modified were marked in red.

Minor point

Point 1: It does not seem to describe the reason why authors focused ferroptosis and/or ferroptosis-related genes methylation to analyze the CM status (or the relationship between CM and ferroptosis) in “Introduction”. Dose the melanoma cells show ferroptosis drastically, or show more drastic than other cancer cells?

Response1: We apologize for our negligence. Excessive ultraviolet (UV) radiation is an important environmental trigger in the pathogenesis of CM [1]. UV from sunlight can lead to excessive intracellular production of reactive oxygen species (ROS), causing oxidative stress damage to cells [2]. Mitochondria, as a major iron-rich and ROS-producing organelle, is considered to be an important site of cell ferroptosis [3], and the accumulation of ROS is one of the characteristics of ferroptosis [4].Thus, the role of ferroptosis-related genes (FRGs) in the development and prognosis of CM has attracted our interest. Recently, there is increasing evidence that FRGs are closely associated with tumor cell proliferation, invasion, metastasis, apoptosis, and tumor therapeutic response in a variety of cancers, including CM. For example, depletion of NEDD4 promotes the Erastin-induced ferroptosis of CM cells [5]. Erastin is a ferroptosis inducer that significantly enhances BRAF inhibitor-induced CM cell death[6]. miR-137 enhanced the therapeutic efficacy by increasing CM ferroptosis [7]. Previous studies have shown that DNA methylation status is more reliable than gene expression in cancer diagnosis and prognosis [8, 9]. DNA methylation can regulate ferroptosis by modulating the transcription of corresponding genes in tumors, affecting various biological behaviors and alterations in signaling pathways in tumor cells [10]. Moreover, due to its high stability, frequency and accessibility, DNA methylation is widely used as a diagnostic, predictive and prognostic biomarker for various cancers [11, 12]. However, to the best of our knowledge, the prognostic performances of FRG DNA methylation in CM remain unclear. Therefore, we focused on the role of FRG DNA methylation in the development and prognosis of CM. According to your kindly suggestions, we have revised the Introduction section.

Point 2: Line 262-263 : “longer” and “lower” are not responding properly.

longer → higher (?)

Response 2: We appreciate the reviewers’ kindly suggestion, and we have changed the “longer” to “higher” to better correspond with “lower”.

Point 3:Insufficient description

In “References” section

Line 375 : 10(9). → 10(9): p.2320.

Line 390 : 13(24). → 13(24): p.6217.

Line 394 : 8. → 8: e44310.

Response 3: We appreciate and accept this suggestion, and we have revised and added the description in the References.

References

  1. Cabrera, R. and F. Recule, Unusual Clinical Presentations of Malignant Melanoma: A Review of Clinical and Histologic Features with Special Emphasis on Dermatoscopic Findings. Am J Clin Dermatol, 2018. 19(Suppl 1): p. 15-23.
  2. Rhee, S.G., Cell signaling. H2O2, a necessary evil for cell signaling. Science, 2006. 312(5782): p. 1882-3.
  3. Gao, M., et al., Role of Mitochondria in Ferroptosis. Mol Cell, 2019. 73(2): p. 354-363 e3.
  4. Perez, M.A., et al., Dietary Lipids Induce Ferroptosis in Caenorhabditiselegans and Human Cancer Cells. Dev Cell, 2020. 54(4): p. 447-454 e4.
  5. Yang, Y., et al., Nedd4 ubiquitylates VDAC2/3 to suppress erastin-induced ferroptosis in melanoma. Nat Commun, 2020. 11(1): p. 433.
  6. Tsoi, J., et al., Multi-stage Differentiation Defines Melanoma Subtypes with Differential Vulnerability to Drug-Induced Iron-Dependent Oxidative Stress. Cancer Cell, 2018. 33(5): p. 890-904 e5.
  7. Luo, M., et al., miR-137 regulates ferroptosis by targeting glutamine transporter SLC1A5 in melanoma. Cell Death Differ, 2018. 25(8): p. 1457-1472.
  8. Paziewska, A., et al., DNA methylation status is more reliable than gene expression at detecting cancer in prostate biopsy. Br J Cancer, 2014. 111(4): p. 781-9.
  9. Hao, X., et al., DNA methylation markers for diagnosis and prognosis of common cancers. Proc Natl Acad Sci U S A, 2017. 114(28): p. 7414-7419.
  10. Zhang, X., et al., Homocysteine induces oxidative stress and ferroptosis of nucleus pulposus via enhancing methylation of GPX4. Free Radic Biol Med, 2020. 160: p. 552-565.
  11. Guo, W., et al., A five-DNA methylation signature act as a novel prognostic biomarker in patients with ovarian serous cystadenocarcinoma. Clin Epigenetics, 2018. 10(1): p. 142.
  12. Aleotti, V., et al., Methylation Markers in Cutaneous Melanoma: Unravelling the Potential Utility of Their Tracking by Liquid Biopsy. Cancers (Basel), 2021. 13(24): 6217.

Reviewer 2 Report

I have enclosed my review as a PDF file 

Round 2

Reviewer 2 Report

The manuscript has been satisfactorily improved  by the authors who has answered to all the raised questions and have reported new data. These latter contribute to give a clearer understanding of the scientific problem.